∂ | **Open Peer Review** | Environmental Microbiology | Research Article

# Ecophysiology and niche differentiation of three genera of polyphosphate-accumulating bacteria in a full-scale wastewater treatment plant

Z. Kondrotaite,[1] J. Petersen,[1] C. Singleton,[1] M. Peces,[1] F. Petriglieri,[1] T. B. N. Jensen,[1] M. Sereika,[1] A. O. H. Daugberg,[1] M. Wagner,[1,2] M. K. D. Dueholm,[1] P. H. Nielsen[1]

**ABSTRACT** Polyphosphate-accumulating organisms (PAOs) are the main bacteria responsible for phosphorus removal and recovery in full-scale wastewater treatment plants (WWTPs). They encompass members of the genera *Candidatus* Accumulibacter, *Azonexus* (formerly *Dechloromonas*), and *Candidatus* Phosphoribacter (formerly *Tetrasphaera*), with most studies focusing on *Ca.* Accumulibacter, primarily using lab-scale enrichment cultures. Although members from the three genera often co-exist in full-scale WWTPs, the metabolic capabilities and traits that determine the niche differentiation of the specific species are still unknown. We retrieved 214 high-quality metagenome-assembled genomes from a full-scale plant with phosphorus removal and examined the polyphosphate-related metabolic pathways using genome-resolved metatranscriptomics in the different process tanks *in situ* and by using short-term incubations *ex situ*. We observed the co-existence of nine uncultured PAO species from the three genera with clear niche differentiation in the utilization of different carbon sources and involvement in the denitrification process. Additionally, we observed several physiological differences among species of the same genus, indicating variations in niche specialization. This suggests that biological P removal and other processes in full-scale WWTPs are carried out by a complex and diverse PAO community that together ensures stable plant performance.

**IMPORTANCE** The current understanding of the ecology and physiology of polyphosphate-accumulating organisms (PAOs) is mostly based on *Candidatus* Accumulibacter, primarily studied in enriched lab-scale studies. Recent taxonomic reclassification revealed that the most studied *Ca.* Accumulibacter species are either not present or present in low abundance in full-scale wastewater treatment plants (WWTPs). This raises concerns that knowledge from lab-scale studies may not apply to species in full-scale plants. Additionally, the indication of a distinct PAO physiology in *Candidatus* Phosphoribacter compared to *Ca.* Accumulibacter and the other abundant PAO *Ca.* Azonexus poses further questions about the accuracy of the current PAO model. Here, we show that in full-scale plant species from *Ca.* Accumulibacter, *Ca.* Azonexus, and *Ca.* Phosphoribacter always co-exist, and they have distinct niche separations in terms of carbon source utilization and the use of electron acceptors. This co-existence and metabolic diversity indicate that a complex microbial community is crucial for efficient phosphorus removal in full-scale WWTPs.

**KEYWORDS** activated sludge, polyphosphate-accumulating organisms, niche differentiation, phosphate recovery, Accumulibacter, Phosphoribacter, *Azonexus*

**Peer Reviewer** Ryuichi Hirota, Hiroshima University, Higashi-Hiroshima, Japan

Address correspondence to P. H. Nielsen, phn@bio.aau.dk.

The authors declare no conflict of interest.

See the funding table on p. 15.

Polyphosphate-accumulating organisms (PAOs) are a group of bacteria used in the enhanced biological phosphorus removal (EBPR) technology to capture phosphate (P) from wastewater to prevent eutrophication and, more recently, to recover P as a fertilizer (1, 2). These bacteria can store large amounts of inorganic P intracellularly as polyphosphate (poly-P) and cycle it through anoxic-oxic feast-famine phases (1). Currently, three bacterial genera comprise the confirmed PAO species: *Candidatus* Accumulibacter, *Azonexus* (formerly *Dechloromonas*), and *Candidatus* Phosphoribacter (formerly *Tetrasphaera*). The filamentous genus *Candidatus* Microthrix is also known to store poly-P but does not show the typical anoxic-oxic poly-P cycling dynamics (3). Several other genera (e.g., *Nitrospira, Halliangium,* and *Thiothrix*) are considered potential PAOs, but their activity has not been confirmed *in situ* (4–6). At the molecular level, the low-affinity P transporter (Pit) may be a determinant for the PAO phenotype, and it is encoded in the genomes of all *in situ* confirmed PAOs (7–9). The use of the Pit transporter may be an advantage to PAOs, as it is driven by the proton motive force and is energetically more efficient than the ATP-dependent high-affinity (PstSCAB) transport system (10). However, the *pit* transporter gene is also widespread among many other bacteria, and its function varies in response to different environmental cues (10), so it is still unknown what specifically determines the PAO phenotype.

*Ca*. Accumulibacter is usually described as the model PAO and has been intensively studied in lab-scale enrichments and a few full-scale studies (7, 11–17). The taxonomy of this lineage has been a subject of some confusion, but recent re-evaluation using a combination of phylogenomic analysis and polyphosphate kinase gene (*ppk*)-based phylogeny has clarified the classification and provided names for 19 distinct species (7). Members of *Ca*. Accumulibacter assimilate organic substrates and degrade poly-P to synthesize polyhydroxyalkanoates (PHAs) under anoxic conditions, while releasing the generated phosphate into the bulk liquid. Under oxic conditions, the stored PHA is used for growth, and P is assimilated to synthesize poly-P. The captured P, as poly-P, is removed from the plant through the surplus sludge discharge (18, 19). The genus largely relies on the uptake and conversion of volatile fatty acids to PHA and uses glycogen as an energy reserve during cycles (1, 7, 16). Recently, it has been proposed that some *Ca*. Accumulibacter species may consume glucose (20, 21), although this has never been confirmed *in situ*. Similarly, some uncultured species belonging to the genus *Azonexus* can cycle P and use acetate and amino acids but not glucose as a carbon (C) source (8). Both *Ca*. Accumulibacter and *Azonexus* are involved in the denitrification process (13, 15–17, 22), but a complete denitrification pathway has only been observed in some *Azonexus* species (1, 8).

The genus *Ca*. Phosphoribacter has a different metabolism compared to *Ca*. Accumulibacter and *Ca.* Azonexus. The understanding of the metabolism of this lineage has been based on studies of the isolated *Tetrasphaera elongata* (23–25). However, a recent genome-based reclassification of this group into different genera has highlighted that *Ca*. Phosphoribacter and not *Tetrasphaera* spp. is abundant in full-scale plants (9, 26). *In situ* studies and metabolic reconstruction have shown that *Ca*. Phosphoribacter is unable to store glycogen and PHA (9, 27, 28) and appears to have different substrate preferences compared to *Ca*. Accumulibacter and *Azonexus*, which can utilize sugars, amino acids, and perhaps acetate as C sources (9). *Ca*. Phosphoribacter has the potential to ferment, which would be an advantage compared to the other PAOs in dynamic environments (9).

Our recent global investigation of microbial communities in wastewater treatment plants (WWTPs), including EBPR plants (29), confirmed the presence and possible coexistence of these three PAO genera worldwide, but the diversity and specific metabolic traits determining their niche differentiation are still poorly understood. As they are all uncultured, previous metabolic studies of PAOs have been performed on enriched lab-scale cultures and may not be representative of the species present in full-scale plants (11, 13, 16, 17). In this study, we used genome-resolved metagenomics and metatranscriptomics to study PAOs in a full-scale EBPR plant. We also applied

short-term lab-scale batch incubations with multiple C sources and electron acceptors to understand their key metabolic traits and niche differentiation. Overall, our study revealed the co-existence of nine abundant PAO species and their niche differentiation, suggesting that biological P removal (and other processes) in full-scale WWTPs is carried out by a complex and diverse PAO community that together ensures stable plant performance.

## MATERIALS AND METHODS

### Sampling in full-scale WWTP and short-term incubations

Sampling was carried out at the Aalborg West (AAW) municipal WWTP, which is designed to remove C, N, and P and has a treatment capacity of 330,000 person equivalents. The plant has an alternating operation with denitrification (DN, anoxic) and nitrification (N, oxic). DN and N cycles last 20–30 minutes and are controlled online based on ammonia and nitrate concentrations. After the secondary settler, the return sludge (RS) is sent to the DN tanks, while a small fraction (ca. 20%) is directed to two anoxic side-stream hydrolysis tanks (SSH), each with a sludge residence time of 27 hours (Fig. 1B). The average sludge residence time of the plant is 22 days. The plant occasionally doses small quantities of iron to improve precipitation of phosphorus and enhance flocculation. During the sampling period, the plant operated efficiently and consistently, meeting effluent standards for total phosphorus (1 mg P/L) and total nitrogen (8 mg N/L). The influent levels to the plant are, on average, 7 mg P/L and 41 mg N/L.

A sample for metagenome sequencing was taken from the aeration tank in AAW WWTP in November 2021 and kept at $-80°C$ prior to processing. Ten additional samples from AAW WWTP were used in parallel for better binning and assembling (Table S1).

Samples for metatranscriptomics were taken from all tanks (DN, N, RS, and SSH) and snap-frozen in liquid $N_2$ and stored at $-80°C$ until further use. In DN and N tanks, samples were taken at three and two different time points within the cycle, respectively. For all samples, five biological replicates were collected.

Short-term incubation experiments were conducted in duplicate to investigate gene expression levels following the addition of various C sources or electron acceptors. Fresh activated sludge was collected from the aeration tank for these experiments. The sludge was collected the day before every incubation experiment and stored at 4°C until used. For P release under anoxic conditions, four different C sources (acetate, glucose, mix of amino acids, and oleic acid, plus control) were evaluated. In a different short-term incubation experiment, four different electron acceptors (oxygen [$O_2$], nitrate [$NO_3^-$], nitrite [$NO_2^-$], and nitrous oxide [$N_2O$]) were evaluated for P uptake after previous anoxic conditions with a mix of C sources. All samples were snap-frozen in liquid $N_2$ and stored at $-80°C$ until further use. Details about the samples and the setup of the incubation experiment can be found in File S1.

### Metagenomics

The DNA was extracted using DNeasy PowerSoil Pro Kit (Qiagen, Germany) following the manufacturer's recommendations. DNA concentration and quality were checked using Agilent TapeStation genomic DNA screen tapes, Qubit 3.0 fluorometer (Thermo Fisher Scientific, MA, USA), and Nanodrop ND1000 (Thermo Fisher Scientific, MA, USA). After size selection, DNA was stored in the TE buffer (pH 8). DNA libraries for Nanopore sequencing were prepared using the SQK-LSK110 Ligation Sequencing kit (Oxford Nanopore Technologies, UK) according to the manufacturer's instructions. The libraries were sequenced with R9.4.1 chemistry flow cells using a GridION sequencing device.

Short read sequencing libraries were prepared using the Illumina DNA Prep kit in combination with IDT UD Indexes Set A according to the manufacturer's instructions. Final libraries were quantified, and insert size was evaluated using the Qubit dsDNA HS assay and a D1000 screentape. Samples were multiplexed using 100 ng of DNA, and the

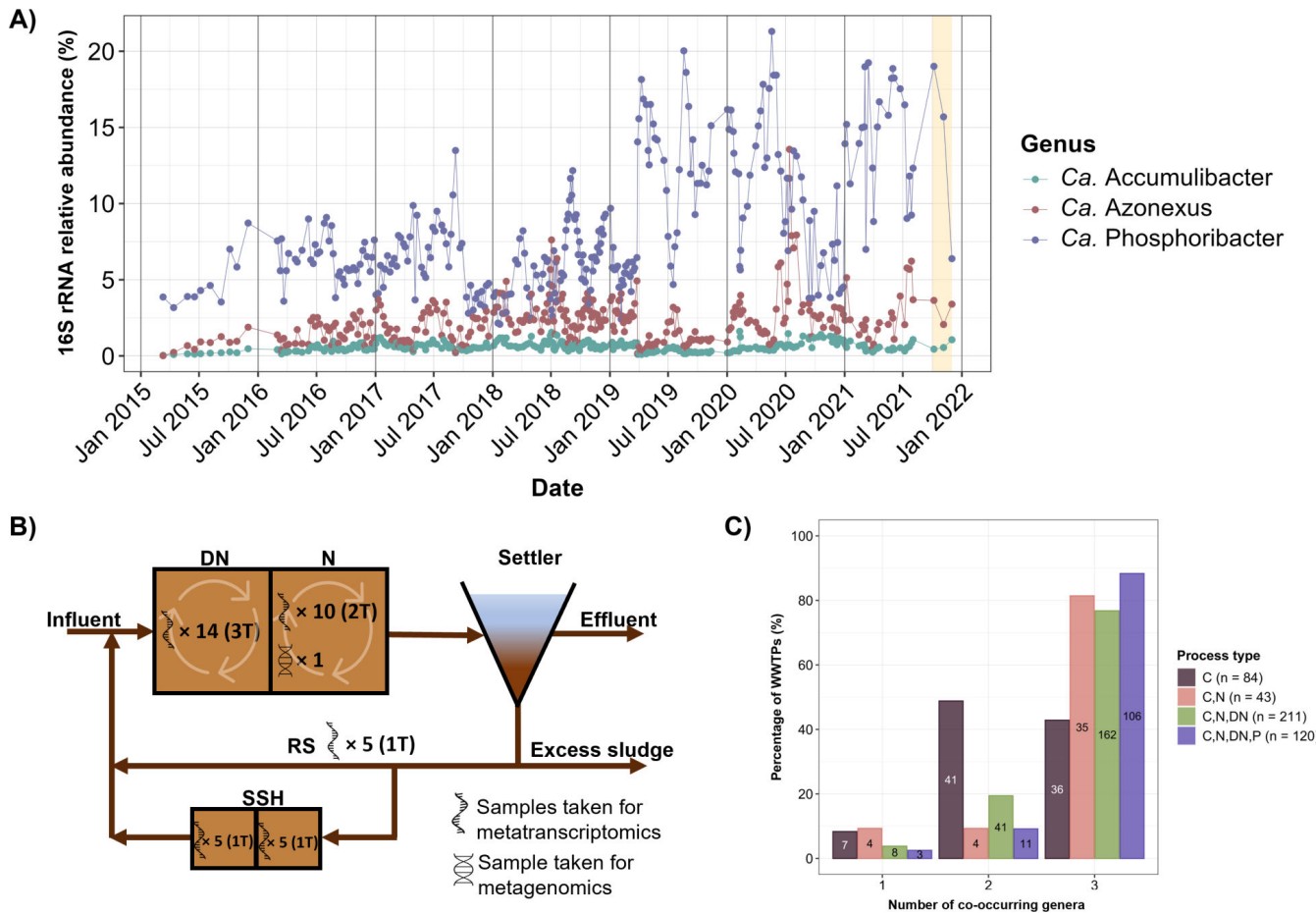

**FIG 1** Relative abundance of three PAO genera and process design of the AAW WWTP. (A) The relative abundance of the three PAO genera as determined by amplicon sequencing (V1–V3) over a 7-year period. The duration of the experiment is identified by the yellow bar. (B) The design of the full-scale AAW WWTP. The numbers represent the number of samples taken in different tanks. Different time points of sampling are identified in brackets as (XT). (DN, denitrification tank [anoxic]; N, nitrification tank [oxic]). The arrows in the DN and N tanks represent mixing within the tank. (C) Co-occurrence of PAO genera in global full-scale WWTPs based on plant design (C, carbon removal; N, nitrification; DN, denitrification; and P, biological P removal).

final pooled library was evaluated in the same way as for the individual libraries. The libraries were sequenced with Illumina NovaSeq 6000 using S4 flow cells.

The sequenced Nanopore reads were assembled using metaFlye version 2.9.1 (30), polished with Racon version 1.5.0 (3× rounds) (31) and Medaka version 1.6.1 (2× rounds) (https://github.com/nanoporetech/medaka), followed by polishing with the generated short Illumina reads using Racon version 1.5.0. Automated contig binning was performed using the ensemble method with Metabat2 version 2.15 (32), MaxBin2 version 2.2.7 (33), Vamb version 3.0.7 (34), and Metabinner version 1.4.3 (35), while Das Tool version 1.1.3 (36) was used to generate the final refined metagenomic bins. To improve genome recovery, coverage values from multiple Illumina read data sets were used as input for the binners.

### Metagenome binning and HQ MAGs database curation

High-quality (HQ) metagenome-assembled genomes (MAGs), including full-length rRNA genes, were determined based on mmlong2 version 0.0.1 (https://github.com/Serka-M/mmlong2). In some cases, the 16S or 23S rRNA genes were not detected by barrnap version 0.9 (https://github.com/tseemann/barrnap). Therefore, Infernal version 1.1.2 (https://github.com/EddyRivasLab/infernal) (arguments: cmscan --cut_ga --rfam --nohmmonly --fmt 2) was used as described in reference 37. Dereplicated bins (95%

ANI) were checked for completeness and contamination using CheckM2 version 0.1.3 and CheckM1 version 1.1.2 (38) –lineage_wf, resulting in 214 HQ MAGs. If a MAG was considered HQ according to the MIMAG standard (>90% completeness and <5% contamination, including rRNA genes) (39) based on completeness and contamination in either CheckM1 or CheckM2, it was kept in the AAW HQ MAG set. Circular contigs, and therefore likely closed genomes, were identified by searching the Flye assembly_info.txt output in the column "circ." for "Y" designation.

The HQ MAGs from the MiDAS genome database from Singleton et al. (37) were combined with the HQ MAGs recovered in this study to make a comprehensive activated sludge database for mapping with metatranscriptomes. The combined MAGs were dereplicated using dRep version 2.3.2 (40) with the CheckM1 completeness and contamination values to remove overlap between the MAG sets and select for the best MAG species representatives after 95% ANI clustering. Multiple sequence alignments of 120 concatenated single-copy proteins, produced by GTDB-Tk version 2.4.0, were used as input for IQ-TREE version 2.0 (41) to create a maximum likelihood PAO tree using the WAG + G model and 1,000-replicate ultrafast-bootstrap analysis. The resulting trees were further examined and rooted in ARB version 7.0 (42), and ITOL version 6 (43) was used for tree visualization, with final aesthetic changes made in Inscape version 1.3.2. Prokka version 1.14.0 (44) "--kingdom Bacteria" was used to call ORFs and provide the coding regions/proteins for all genomes. The protein amino acid FASTA file was compared with the KofamKOALA database version 1.3.0 (45) to obtain annotation and functional annotation information. In addition, the analyzed HQ MAGs were uploaded to "MicroScope Genome Annotation and Analysis Platform" (46) to cross-validate KO annotations found using KofamKOALA. Gene localization was analyzed using psortB (47).

## Metatranscriptomics

RNA was extracted using the RNeasy PowerMicrobiome kit (Qiagen, Germany) following the manufacturer's instructions. Extracted RNA was treated with DNase using the TURBO DNA-free kit (Thermo Fisher Scientific, Lithuania) and cleaned with RNAclean XP beads. RNA concentration and quality were checked using Qubit 3.0 fluorometer (Thermo Fisher Scientific, MA, USA) and Agilent TapeStation RNA ScreenTapes (File S3). Library preparation was performed using NEBNext rRNA Depletion Kit (Bacteria) with RNA Sample Purification Beads, followed by NEBNext Ultra II Directional RNA library preparation (New England Biolabs), based on the manufacturer's recommendations. Library concentration and quality control were performed using Qubit 3.0 fluorometer (Thermo Fisher Scientific, MA, USA) and Agilent TapeStation High Sensitivity D1000 ScreenTapes (File S3). The samples were sequenced on a NovaSeq 6000 using a 300-cycle version 1.5 S4 flow cell and reagents (kit 20028312). More details on sequencing, read quality control, and filtering can be found in File S1.

The filtered reads were mapped to the coding sequences (CDSs) of the HQ MAG database using RSEM version 1.3.3 (48). This was done using the rsem-prepared-reference command with the "-bowtie2" flag, followed by the rsem-calculate-expression command with the following flags: "-paired-end, -bowtie2, -bowtie2-sensitivity-level sensitive." This generated expected read counts, which were used to calculate the transcript per million (TPM) value for each CDS in the HQ MAG database for each sample. The TPM values are a measure of the relative expression of a given CDS compared to all other CDSs in a sample. The formula used for the TPM calculations was

TPM = (read counts of the gene/effective length of the gene)/[sum of (read counts/ effective length) for all genes in the sample] × 1,000,000

## RESULTS AND DISCUSSION

### The microbial community of a full-scale EBPR plant

Microbial community composition plays a crucial role in the stability and efficiency of EBPR processes in WWTPs. We have followed the microbial community composition in

several Danish full-scale EBPR plants with weekly sampling and 16S rRNA gene amplicon analyses for more than 5 years (49). Aalborg West was selected for further studies due to its stable performance for many years, and all three PAO genera were present, with *Ca*. Phosphoribacter as the most abundant, accounting for up to 21% of the 16S rRNA gene relative abundance (Fig. 1). The abundances of the different PAOs varied over time, likely influenced by factors such as process design, wastewater type, plant operation, immigration via wastewater, and microbial interactions. We do not, however, have any specific explanations for the observed dynamics (49).

We retrieved 214 HQ MAGs from AAW, and in combination with other HQ MAGs from Danish WWTPs (37), we created a HQ MAG database with 692 species representatives covering 378 different bacterial genera in 29 phyla (File S2). The most abundant species-representative MAGs belonged to *Ca*. Phosphoribacter, *Nitrospira,* and *Azonexus* genera, with relative abundances of 3.13%, 1.79%, and 1.58%, respectively (Fig. S1). *Ca*. Accumulibacter species were less abundant (up to 0.06%). This is consistent with the results obtained from amplicon sequencing (Fig. 1A) and aligns with the common pattern observed globally, where *Ca*. Phosphoribacter is typically more abundant than the other two PAO genera (29). The overall community composition confirmed the general core community found in previous studies of full-scale WWTPs (29, 50).

Metatranscriptomic data showed that the HQ MAGs representing bacteria with the greatest transcript expression belonged to the nitrifier *Nitrospira* and to the two PAOs, *Ca*. Phosphoribacter and *Ca*. Azonexus (Fig. S2), which correlated with the MAG abundances determined from the metagenomes (Fig. S1). Overall, expression levels for most species were very similar in DN and N tanks, while the highest variations were observed in the anoxic SSH (Fig. S2), showing functional and metabolic differences within the microbial community.

The coexistence of all three PAO genera in full-scale WWTPs has previously been observed in Danish EBPR plants (51), and our re-evaluation of the global WWTPs based on 16S rRNA gene data from the MiDAS 4 study of >700 WWTPs (29) showed that coexistence is a recurrent feature. We found that all three PAO genera were commonly found together, particularly in WWTPs with process design for N and P removal (Fig. 1C).

In AAW, we found 15 species-representative HQ MAGs of PAOs (3 *Ca*. Accumulibacter, 6 *Ca*. Phosphoribacter, and 6 *Ca*. Azonexus, Fig. 2), and of these, *Ca*. Phosphoribacter hodrii, *Ca*. Phosphoribacter baldrii, and *Ca*. Azonexus phosphoritrophus were the most abundant (0.9%–3.1% relative abundance). These species were also the most abundant ones within the two genera in previous amplicon studies of Danish WWTPs (8, 9). We identified several novel species and proposed new names for all 8, such as *Ca*. P. hoenirii (Fig. 2; File S1; Tables S3 to S10). Among the *Ca*. Accumulibacter species, *Ca*. Accumulibacter proximus (clade IID) was the most abundant (0.06%). Interestingly, none of them were species found in enriched bioreactor studies, which mostly represent *Ca*. Accumulibacter clades IA, IIA, and IIC (12, 16, 17, 35).

## Niche differentiation of PAOs

Recent metabolic reconstruction of members of the three PAO genera revealed differences in the metabolic potential and plausible niche differentiation (7–9, 21). However, the exact mechanism used for the poly-P accumulation, preferences in C sources, storage polymers, and involvement in the denitrification process are not clear, even for well-studied *Ca*. Accumulibacter species (12, 16, 17). To determine these mechanisms in known PAOs, we evaluated the expression levels of specific genes (Fig. 3; Table S2; File S4), both *in situ,* by performing short-term laboratory incubations where the AAW microbial community was spiked with individual C sources and electron acceptors (Fig. 4). In this study, we focused on the nine most abundant co-existing PAO MAGs (Fig. 3), with *Ca*. Accumulibacter proximus, *Ca*. Azonexus phosphoritrophus, and *Ca*. Phosphoribacter hodrii having the highest expression within each genus (Fig. 2).

In general, we observed lower expression levels in the anoxic SSH tanks compared to DN and N tanks for all nine PAO species (Fig. S3), in contrast to enriched lab-scale

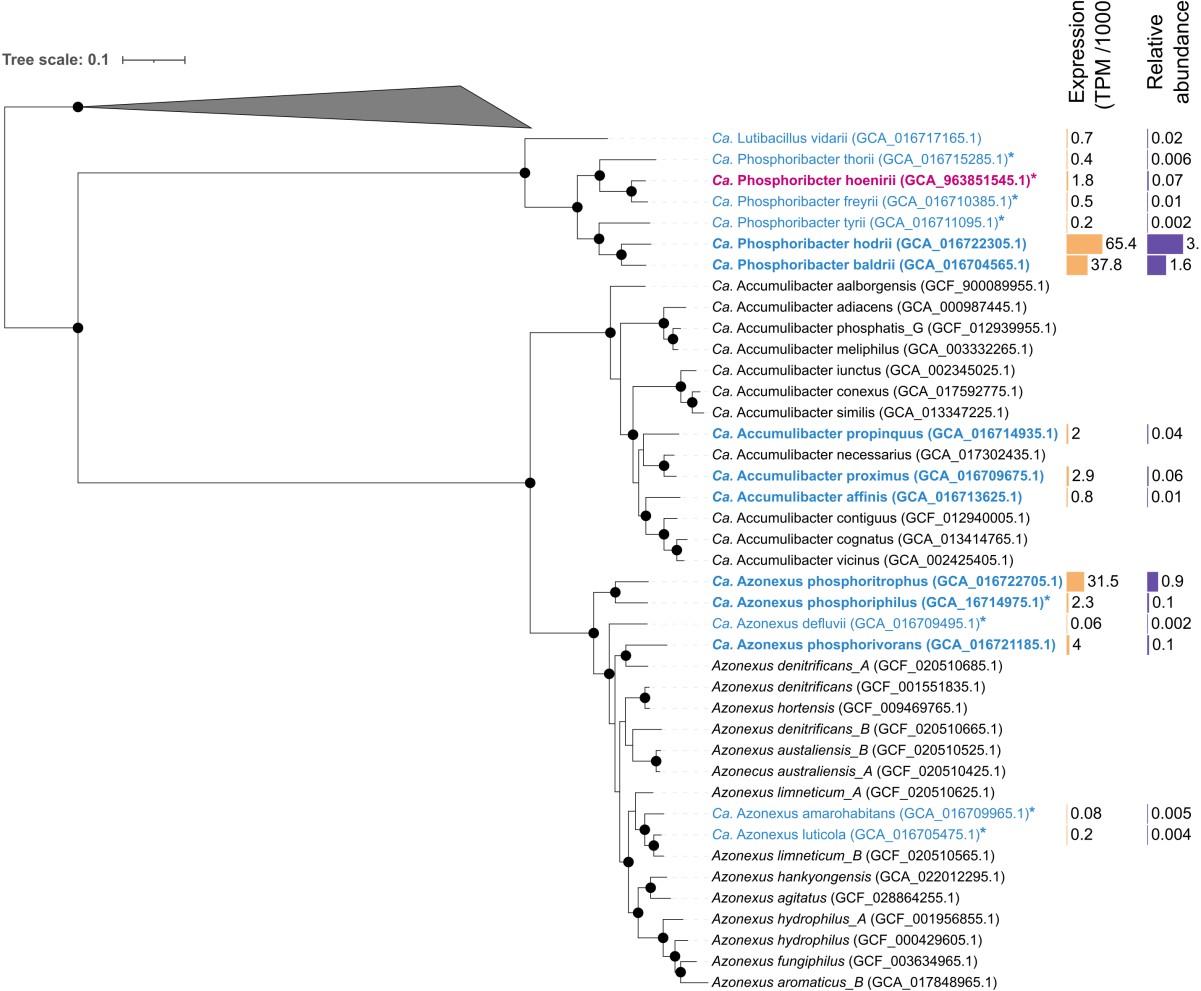

**FIG 2** Phylogenetic tree showing the diversity and phylogenetic classification of three PAO genera. The HQ MAGs representing *Ca*. Accumulibacter, *Azonexus*, and *Ca*. Phosphoribacter species are shown. The tree is based on the concatenated alignment of 120 single-copy marker gene proteins using GTDB-Tk version 2.4.0. Bootstraps are shown with the value of ≥95%. HQ MAGs recovered from this study are shown in pink. HQ MAGs from reference 37 are shown in blue. HQ MAGs used for further analysis are in bold. MAG expression and relative abundance are shown with the bar plot. MAGs from the Chloroflexota phylum were chosen as the outgroup. Novel species named in this study are indicated by *.

experiments, where the expression levels of *Ca*. Accumulibacter phosphatis UW-1 were very similar between the anoxic and oxic phases (16). The incubations with the four different C sources (acetate, glucose, a mix of amino acids, or oleic acid) showed the highest P release with acetate (Fig. 4A). Glucose appeared to be the second most favorable substrate for P release, followed by the mix of amino acids (Fig. 4A). The incubations with different electron acceptors ($O_2$, $NO_3^-$, $NO_2^-$, and $N_2O$) showed that P uptake was only observed when the microbial community was aerated with $O_2$ or supplied with $NO_3^-$ in the absence of $O_2$. No P uptake was detected with $NO_2^-$ or $N_2O$ as electron acceptors (Fig. 4B).

### Niche separation in consumption of C sources

The utilization of different C sources by PAOs is a topic of ongoing discussion, particularly the potential use of glucose by *Ca*. Accumulibacter and the favorable C source of the newly characterized *Ca*. Phosphoribacter. Based on lab-scale experiments (52, 53) and metabolic prediction from our and other studies (Fig. 4 [8, 17]), all species of *Ca*. Accumulibacter and *Ca*. Azonexus appeared capable of anaerobic uptake of acetate, amino acids, and oleic acids. This potential was also supported by our

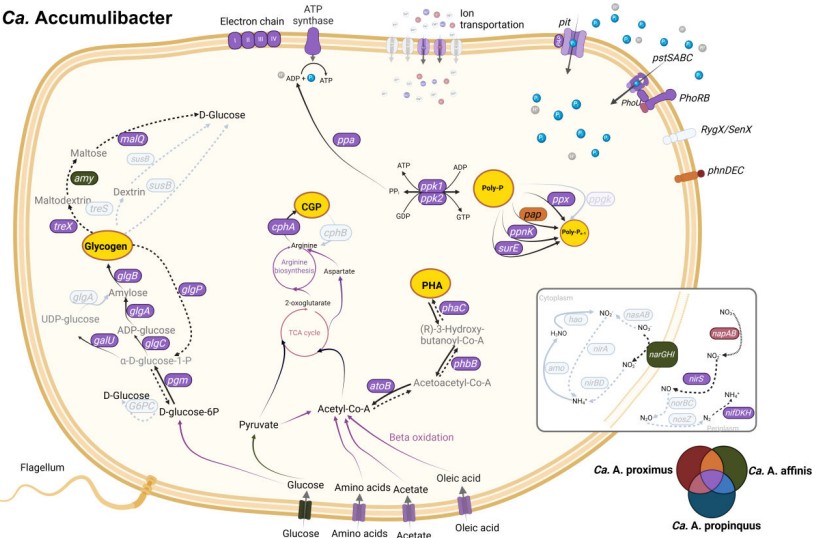

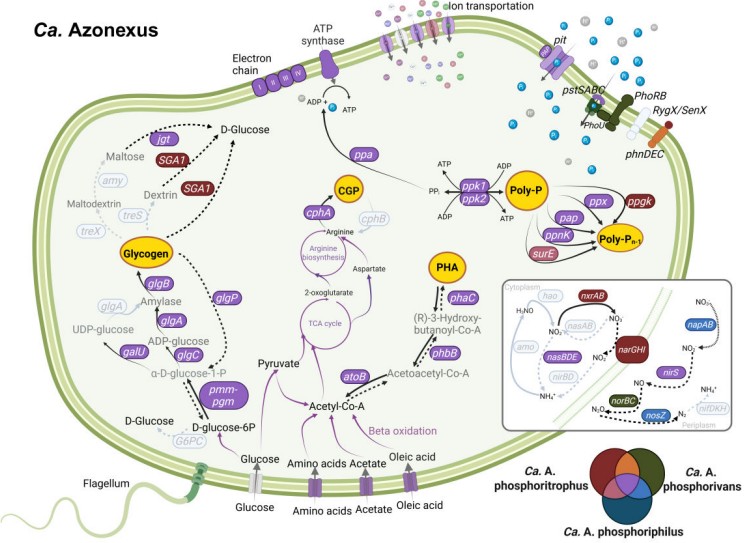

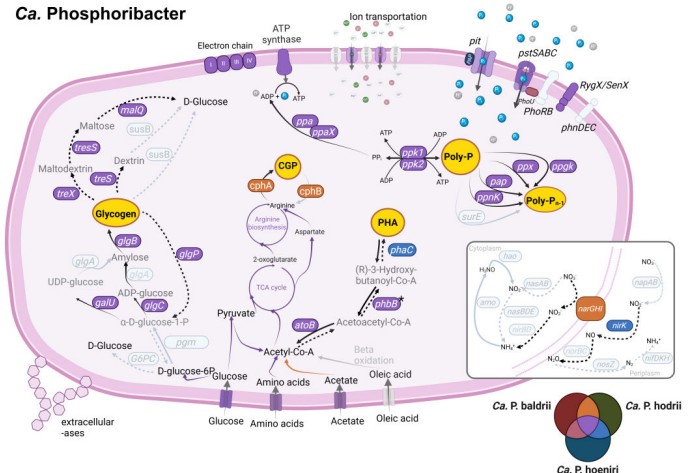

**FIG 3** Metabolic models of PAOs from the full-scale EBPR plant. Colors indicate the presence of genes in different species (*Ca*. A. proximus, *Ca*. A. propinquus, *Ca*. A. affinis, *Ca*. A. phosphoritrophus, *Ca*. A. phosphorivorans, *Ca*. A. phosphoriphilus, *Ca*. P. baldrii, *Ca*. P. hodrii, and *Ca*. P. hoeniirii). Gray color

Fig 3 (Continued)

indicates that the gene/pathway is not present in any of the species of a specific genus. For specific genes in the pathways, see Data S4. Continuous arrows indicate that the reaction happens under oxic conditions, and dashed arrows indicate that the reaction happens under anoxic conditions. Metabolic model reconstruction is based on the KofamKOALA and the MicroScope platform. If the gene was annotated only by one annotation software, it is indicated with an asterisk (*) next to the gene name.

metatranscriptomic data, where genes related to the utilization of the C sources were highly expressed in the two genera (Fig. S5 to S7). The total expression (i.e., total expression of all MAG genes) of *Ca.* Accumulibacter and *Ca.* Azonexus was stable with the addition of all C substrates except acetate, which led to a decreased expression (Fig. S4). Acetate is usually consumed within 1 hour after addition in short-term incubation experiments (13, 16), during which *Ca.* Accumulibacter and *Ca.* Azonexus likely reached their maximum capacity for PHA production, which may be reflected in a significant decrease in total expression levels.

Utilization of glucose by members of *Ca.* Accumulibacter was proposed in two recent studies (20, 21) but has not yet been confirmed *in situ*. In our study, only *Ca.* A. affinis possessed the full gene set of the glucose transporter (*ptsGHI*), while the other two species possessed only two (*ptsHI*) out of three genes (File S4), which still could be used for other sugar transportation (54), as the *ptsG* makes the transporter system glucose specific (55). However, the expression of the *ptsGHI* genes was very low in *Ca.* A. affinis in the full-scale plant, and the addition of glucose in short-term incubations did not influence the expression levels of the genes (Fig. S5 and S6). Yet, the low expression of glucose transporter genes may be a result of low general expression, and further studies are required to confirm this result. However, the presence of the *ptsG* in *Ca.* A. affinis indicates that glucose utilization by *Ca.* Accumulibacter is species dependent rather than a general trait in full-scale plants. Therefore, glucose utilization may vary depending on the species composition, highlighting that more *in situ* studies in more full-scale plants are necessary.

The recent reclassification of *Tetrasphaera* into *Ca.* Phosphoribacter and several other genera raises uncertainty about the capabilities of *Ca.* Phosphoribacter to utilize different C sources, as no representative pure cultures are available. The potential use of glucose, acetate, and amino acids was predicted by Singleton et al. (37) and also identified in this study, yet only genes related to glucose and amino acid metabolism were highly expressed (Fig. S8 and S9). The monosaccharide glucose/mannose transporter (*malK*) was highly expressed in *Ca.* Phosphoribacter both in the full-scale plant and short-term incubations, supporting their utilization of glucose (9). Some members of *Tetrasphaera* may use oleic acid as a C source (56), but it seems not to be the case for *Ca.* Phosphoribacter. None of the species had the *fadL* gene required for long-chain fatty acid transport or the complete genomic potential for beta-oxidation (File S4). Additionally, when the activated sludge was spiked with oleic acid, it did not affect their general gene expression level (Fig. S4).

Fermentation is an important trait of *Ca.* Phosphoribacter, enhancing its survival in the competitive and dynamic environment of WWTPs with the EBPR process. Almost all the genes in relation to fermentation were highly expressed in SSH tanks, with the highest expression in *Ca.* P. baldrii and *Ca.* P. hodrii (Fig. S8), except for alanine dehydrogenase (*ald*). Furthermore, the pyruvate ferredoxin oxidoreductase (*porAB;* anaerobic metabolism) and cytochrome bd ubiquinol oxidase (*cydAB;* aerobic metabolism, anoxic conditions) genes also had higher expression levels (Fig. S8 and S9). This confirms the flexibility of *Ca.* Phosphoribacter to grow under both anoxic and oxic conditions (57–59), providing it with an advantage in comparison to the other two PAO genera.

Another clear niche difference between *Ca.* Phosphoribacter and the other two PAO genera was observed in their potential for extracellular enzymes. *Ca.* Accumulibacter and *Ca.* Azonexus showed limited potential for the production of extracellular enzymes (File S4), indicating a lifestyle mainly relying on soluble, easily consumable substrates. They

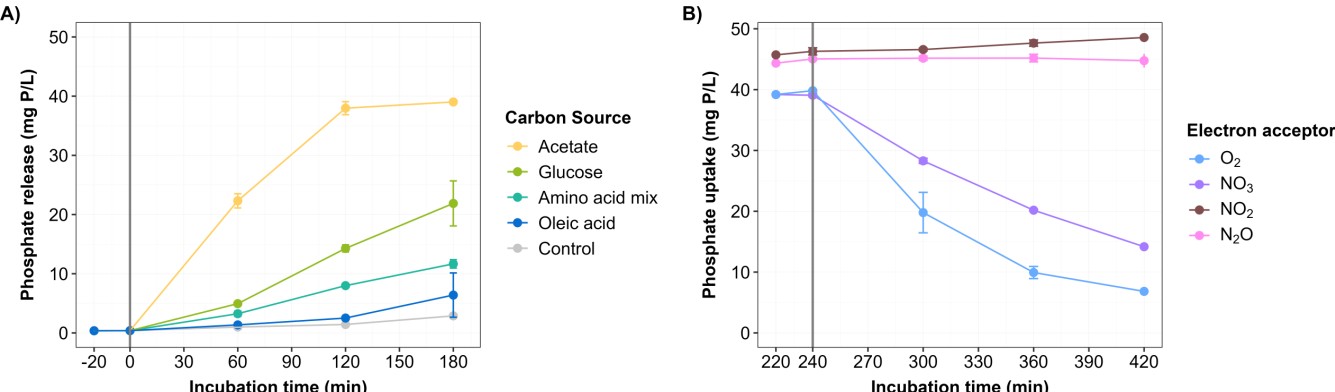

**FIG 4** Effect of C sources and electron acceptors on P release and P uptake. Bulk P concentration measured during the (A) anaerobic P release with four different C sources, and (B) anoxic/oxic P-uptake with four different electron acceptors after previous substrate uptake under anoxic conditions. Short-term experiments with fresh activated sludge from the AAW WWTP (SS approximately 4 g SS/L).

primarily encoded genes for flagella and pili assembly, and in *Ca*. Azonexus, the highest expressed extracellular enzymes were related to the Type IV secretion system and pilus assembly. All *Ca*. Phosphoribacter members lacked genes for flagella and pili assembly but had many genes encoding extracellular peptidases and amylases, indicating the potential to digest macromolecules. The highest expression of genes encoding extracellular enzymes in *Ca*. Phosphoribacter was the *ald* gene (Fig. S12), encoding the alanine dehydrogenase enzyme catalyzing a reversible conversion of L-alanine to pyruvate (60). Overall, most of the extracellular enzymes had low expression levels in all three PAO genera (Fig. S10–S12).

## *Poly-P accumulation*

For poly-P accumulation, the uptake of P is crucial and can be facilitated by two distinct transporters encoded by *pit* and *pstSCAB* genes (10). Yet, only multiple copies of the *pit* gene have been consistently identified in all confirmed PAOs (Fig. 3; File S4) (7–9). On the contrary, the *pstSCAB* transport system also requires the *phoURB* genes that regulate its expression (10). All *pstSCAB* + *phoURB* genes were fully present in *Ca*. Accumulibacter species and *Ca*. Azonexus phosphorivans, but not in other *Ca*. Azonexus species (8). *Ca*. Phosphoribacter had all *pstSCAB* genes but lacked the *phoURB* genes, suggesting different regulation of the transporters. *Ca*. Phosphoribacter had a low expression of the *pstSCAB* genes in comparison to the expression of the *pit* gene (Fig. 5). The results showed that when PAOs had both transport systems and *phoURB* genes, both were expressed at similar levels (Fig. 5), except in *Ca*. A. affinis, where *pstSCAB* + *phoURB* had higher expression than the *pit*. The expression of both transport systems may enable *Ca*. Accumulibacter to accumulate more poly-P than other PAOs within the same time period. This, combined with their larger cell size, allows for the storage of greater overall amounts of poly-P (61).

Variations in *pit* gene expression under different conditions in PAOs are of significant interest due to its presence in all confirmed PAOs. Notably, we observed different expression patterns of this gene in *Ca*. Phosphoribacter compared to *Ca*. Accumulibacter and *Ca*. Azonexus (Fig. 5; Fig. S13 and S14). This variation was also evident in short-term incubations. Additionally, higher variations in *pit* gene expression were observed with the addition of different C sources in comparison to the addition of different electron acceptors, which suggests that the C sources had a greater influence on the expression of this gene than different electron acceptors (Fig. S13 and S14). The observed differences in *pit* gene expression may be due to different gene homologs in *Ca*. Accumulibacter and *Ca*. Azonexus than in *Ca*. Phosphoribacter.

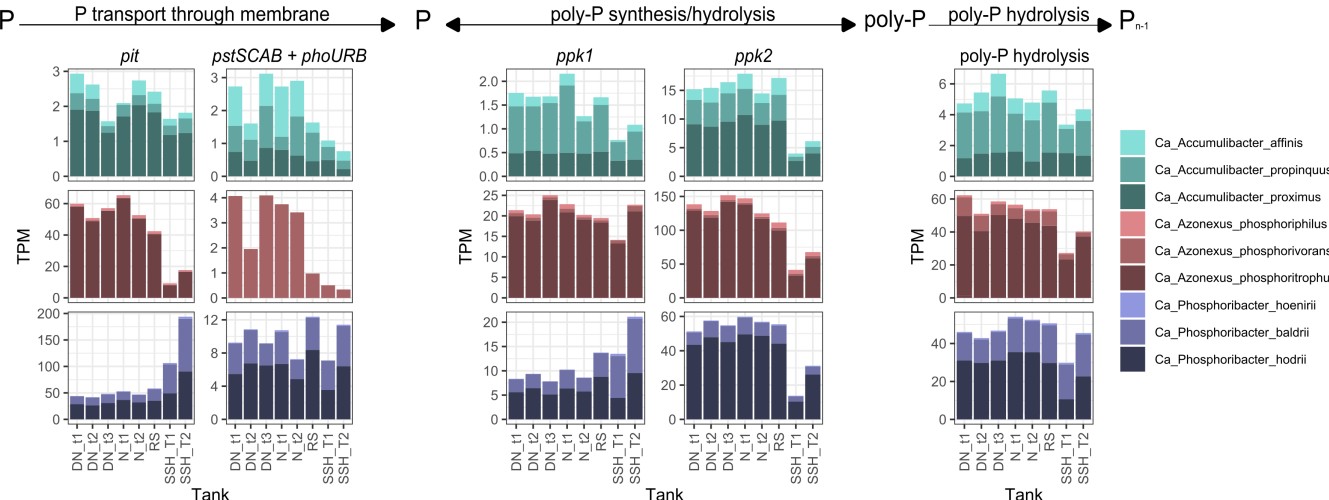

**FIG 5** Expression levels of the genes related to poly-P metabolism. The graphs show expression levels of the genes related to P transport, P synthesis/hydrolysis of PAO species in different stages of full-scale WWTP, and in different times of the cycles in DN and N tanks. tX, different time points; T1/2, different SSH tanks. The list of genes grouped as poly-P hydrolysis genes can be found in Table S2.

The *ppk1* and *ppk2* genes are directly involved in poly-P synthesis and degradation. It is proposed that the *ppk1* gene is related to poly-P synthesis and *ppk2* to poly-P degradation (62, 63). Similar expression levels of the *ppk1* gene were observed in all oxic and anoxic tanks of the full-scale WWTP for *Ca*. Accumulibacter and *Ca*. Azonexus species, while the expression in *Ca*. Phosphoribacter increased in the anoxic SSH tanks (Fig. 5). The *ppk2* could provide additional energy under anaerobic conditions by regenerating GTP and ATP (64). The expression levels of the *ppk2* gene were similar in all tanks, except in SSH, where expression levels decreased in all species. In addition to *ppk2*, there are six other genes (*ppa, pap, ppaX, ppx, ppnk,* and *ppgk;* Table S2) related to poly-P hydrolysis (65), and they were more expressed in DN, N, and RS tanks compared to SSH tanks (Fig. 5). The inorganic pyrophosphatase (*ppa*) gene, which is part of *ppk1/ppk2* poly-P hydrolysis, had the highest expression in comparison to other poly-P hydrolysis genes. Overall, our results showed that the *ppk1* gene was consistently expressed across different conditions, while the expression of the *ppk2* gene was more dependent on the environmental conditions, with the *ppk1/ppk2-ppa* pathway likely being the primary pathway for poly-P degradation.

## Other storage compounds

PAOs from *Ca*. Accumulibacter and *Ca*. Azonexus use glycogen and PHA as storage polymers (7, 8). However, the presence of storage polymers in *Ca*. Phosphoribacter remains uncertain, as none of the previously mentioned storage polymers have been detected *in situ* (9), and it is doubtful if they encode genes for glycogen formation (Fig. 3; File S4).

For glycogen accumulation in *Ca*. Accumulibacter and *Ca*. Azonexus, *glgABC* and *glgP* genes were highest expressed in comparison to other genes possibly involved in glycogen metabolism (Fig. 3); thus, we decided to focus only on these genes. For *Ca*. Accumulibacter, the *glgABC* genes had higher expression in SSH tanks, and the *glgP* had higher expression in DN, N, and RS tanks. This contrasts the findings of Wang et al. (35), where minimal changes in glycogen and PHA metabolism of *Ca*. Accumulibacter species were observed between the anoxic and oxic phases in lab-scale enrichments. Based on our metabolic prediction, *Ca*. Phosphoribacter lacked the *glgA* gene necessary for glycogen metabolism (Fig. 3), which agrees with the previous observations by Singleton et al. (37), where glycogen was not detected *in situ*.

Although PHA was not detected *in situ* in *Ca*. Phosphoribacter and the full set of *phaABC* genes was not identified in the previous study by Singleton et al. (37), we identified all the *pha* genes in *Ca*. P. baldrii and *Ca*. P. hodrii. Notably, the *phaB* gene was exclusively annotated using the MicroScope platform (46). Thus, the apparent presence of the genes required for PHA storage varied depending on the annotation tools used. Expression of the *phaABC* genes in *Ca*. Phosphoribacter suggests that it could potentially store PHA at low levels, or it produces a different type of PHA that has different detectable ranges *in situ* (66). Unlike *Ca*. Phosphoribacter, the *phaABC* expression was lower in *Ca*. Accumulibacter and *Ca*. Azonexus in the SSH tanks.

Cyanophycin is a newly proposed potential storage polymer for *Ca*. Phosphoribacter (9), and potentially also for the other two PAO genera, as they also possess the genes associated with its biosynthesis (Fig. 3). Use of this storage polymer instead of glycogen was discussed previously (37, 67, 68), but its importance *in situ* is unknown. The *cphA* gene related to the cyanophycin metabolism showed very low expression in *Ca*. Accumulibacter and *Ca*. Phosphoribacter (Fig. 6). However, the potential of cyanophycin as a storage polymer among PAOs needs to be studied in more detail.

## Denitrification

The involvement of PAOs in denitrification is of interest for two reasons: some species may be important for partial or full denitrification in the plants; and P uptake may take place in the DN tank using $NO_3^-$ or $NO_2^-$ as electron acceptor in the absence of oxygen. PAOs capable of taking up P using nitrate are usually referred to as denitrifying PAOs (DPAOs) (11, 69–71).

The clear niche separation of the PAOs present in full-scale plants can be noted by their ability to participate in the denitrification process at the genus and species levels (Fig. 3). Among the three PAO genera, only one species, *Ca*. A. phosphorivorans, had the genomic potential to carry out full denitrification, while all species from *Ca*. Phosphoribacter were the least involved in this process as they encode only the nitrate reductase (Nar). The potential for *Ca*. Accumulibacter to participate in the denitrification, as observed in this study, differs from that of species enriched in lab-scale experiments (16, 52, 71).

The presence of denitrification genes in PAOs gives them an advantage in phosphorus uptake under anoxic conditions, when either $NO_3^-$ or $NO_2^-$ is available. However, the rate of phosphorus uptake in the absence of oxygen is lower compared to when oxygen is present (11, 69–72). Although phosphorus uptake has previously been described with both $NO_3^-$ and $NO_2^-$, in our study, phosphorus uptake occurred with $NO_3^-$ but not when $NO_2^-$ or $N_2O$ were used as electron acceptors (Fig. 4B). P uptake inhibition with $NO_2^-$ as an electron acceptor (Fig. 4B) could be due to the high concentration of $NO_2^-$ used (72) and agrees with previous observations that even small amounts of $NO_2^-$ inhibit aerobic/anoxic poly-P cycling and denitrification (71, 73).

Except for *Ca*. P. hoenirii, species from the three analyzed PAO genera had the capability to reduce $NO_3^-$ to $NO_2^-$ (Fig. 3). *Ca*. A. affinis, *Ca*. A. phosphoritrophus, *Ca*. P. baldrii, and *Ca*. P. hodrii encoded the respiratory nitrate reductase (NarGHI), whereas the remaining species had the periplasmic reductase (NapAB). The key distinction between these two reductases is their capacity to generate energy in the form of ATP. Furthermore, the presence of *narGHI* genes is thought to confer an advantage in dynamic conditions due to their activity under anoxic conditions, whereas *napAB* genes are typically not involved in anaerobic respiration (7, 8). Previously, it was proposed that with the addition of $NO_3^-$, P uptake could be carried out by the PAOs that possess *narGHI* genes and not *napAB* genes, as only the NarGHI enzyme produces sufficient ATP to support bacterial growth and other processes (74–76). Although the expression levels and the exact use of the genes did not correlate, the addition of $NO_3^-$ in the absence of oxygen resulted in an increase in gene expression levels for most of the genes analyzed (specifically related to poly-P transport, synthesis, and hydrolysis) in all three species, with the highest increase in *Ca*. Accumulibacter and *Ca*. Azonexus species (Fig. 7; Fig. S14

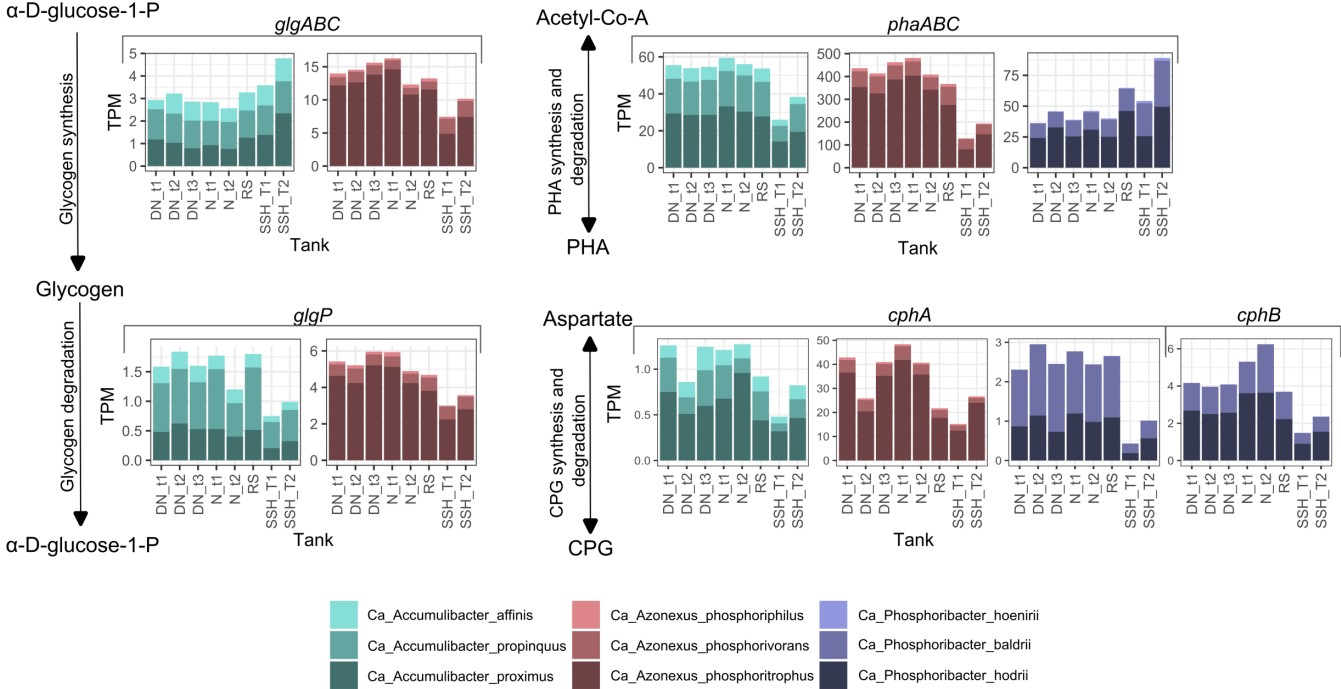

**FIG 6** Expression levels of the genes related to storage polymers. The graphs show expression levels of the genes related to the metabolism of three storage polymers: glycogen, PHA, and cyanophycin of PAO species in different stages of full-scale WWTP and at different times of the cycles in DN and N tanks. tX, different time points; T1/2, different SSH tanks.

to S17), regardless of the presence of *narGHI* or *napAB* genes. Therefore, all the studied species were potential DPAOs, except *Ca*. P. hoenirii, which lacked the *narGHI/napAB* genes (Fig. 3). Thus, the main difference in P uptake with the addition of $NO_3^-$ is most likely due to the lower energy production in the presence of $NO_3^-$ compared to $O_2$.

## Ecological perspective

Our study provides clear evidence of the coexistence of several PAO species within the same full-scale EBPR plant as indicated by the global amplicon survey, highlighting the differences in niche separation not only between different PAO genera but also between species within the same genus. Distinct niche differentiation was observed for the *Ca*. Phosphoribacter, compared to the other two genera. *Ca*. Phosphoribacter appears to be more adapted to survive and thrive under anoxic conditions. Its growth is supported by the ability to ferment, using distinct C sources, particularly in its ability to utilize glucose; however, it is unable to utilize oleic acid. *Ca*. A. affinis showed a potential for glucose utilization, distinguishing itself from other PAO species of *Ca*. Accumulibacter. Overall, *Ca*. Accumulibacter and *Ca*. Azonexus displayed similar preferences for C sources. The observed metabolic similarities between *Ca*. Accumulibacter and *Ca*. Azonexus, along with the distinctiveness of *Ca*. Phosphoribacter, may be attributed to their evolutionary relationship (Fig. S1).

Another clear separation was observed in the differences within the genes encoding extracellular proteins and enzymes. While *Ca*. Accumulibacter and *Ca*. Azonexus had most of the genes related to mobility, *Ca*. Phosphoribacter encoded genes related to the extracellular degradation of macromolecules. At the species level within the three genera, differences were particularly noted for genes related to P transport and the denitrification process.

Major differences were observed regarding P transport of the three genera, specifically for *Ca*. Accumulibacter, which was the only genus with all species possessing both Pit and PstSCAB transport systems. The observed results suggested that *Ca*. Azonexus

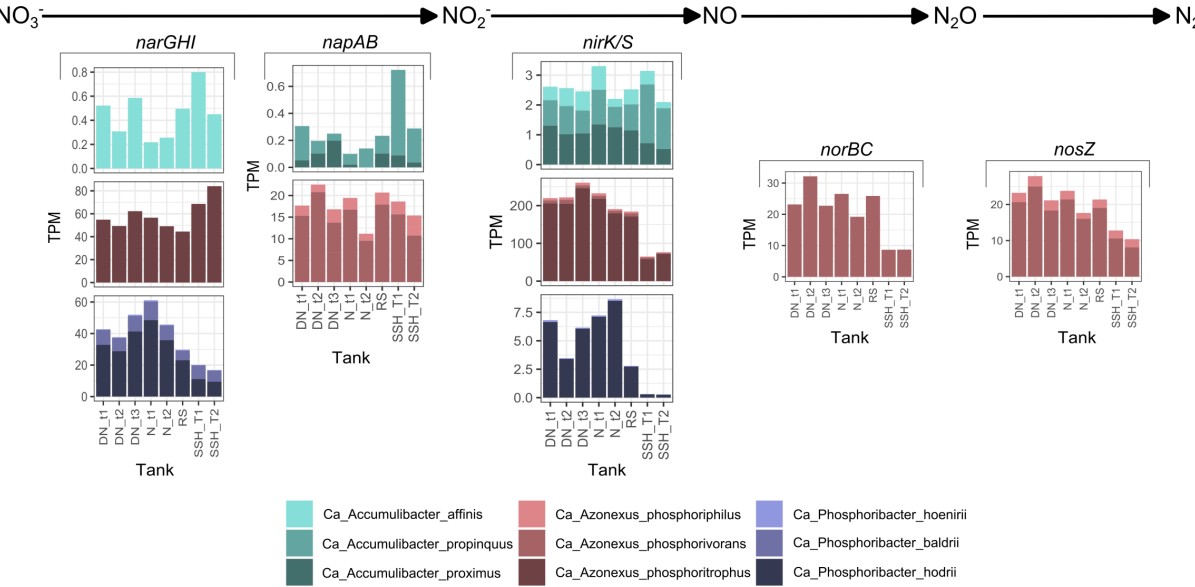

**FIG 7** Expression levels of the denitrification genes. The graphs show expression levels of the denitrification genes of PAO species in different stages of full-scale WWTP and at different times of the cycles in DN and N tanks. TX, different time points; T1/2, different SSH tanks.

and *Ca*. Phosphoribacter potentially used only the Pit transporter, while *Ca*. Accumulibacter used both Pit and PstSCAB + phoURB for P transportation, which could be an advantage for the latter genus. Yet, to fully understand the use of *pit* genes across PAOs, the presence of different gene homologs and their usage in PAOs should be studied in more detail. Moreover, the addition of $O_2$ during the incubations resulted in an increase in the expression of *pstSCAB + phoURB* genes. This could suggest that *Ca*. Accumulibacter is taking up more P under oxic conditions than the other two PAOs.

Another major difference was in *Ca*. Azonexus species and their involvement in the denitrification process. Only *Ca*. Azonexus phosphorivorans could perform full denitrification and was highly involved in P and N removal from wastewater. On the other hand, *Ca*. Accumulibacter species had the potential to reduce $NO_3^-$ to $NO_2^-$ and then to NO. Only *Ca*. P. baldrii and hodrii could potentially reduce $NO_3^-$ to $NO_2^-$ and use $NO_3^-$ as an electron acceptor for poly-P accumulation, leaving only *Ca*. P. hoenirii unable to use $NO_3^-$ as an electron acceptor for poly-P accumulation.

These and other differences could facilitate the coexistence of all PAOs in the same plant and their contribution to nutrient removal. It is also important to note that, while studies using enriched bioreactor cultures provide valuable insights into bacterial metabolic potential, our research focuses on the species that are present and active in full-scale WWTPs, and these species often exhibit different behavior compared to those enriched in bioreactor-based studies.

## ACKNOWLEDGMENTS

We thank all the people who were involved in the Glomicave project. Special thanks to Susanne Bielidt for her invaluable assistance throughout the research process in the lab.

This study was funded by the Glomicave project (Grant 952908 to P.H.N.) and the Villum Foundation (Dark Matter, grant 13351 to P.H.N.).

## AUTHOR AFFILIATIONS

[1]Center for Microbial Communities, Department of Chemistry and Bioscience, Aalborg University, Aalborg, Denmark
[2]Division of Microbial Ecology, Center for Microbiology and Environmental Systems Science, University of Vienna, Vienna, Austria

## AUTHOR ORCIDs

Z. Kondrotaite http://orcid.org/0000-0003-2798-6060

J. Petersen http://orcid.org/0000-0002-1960-2269

C. Singleton http://orcid.org/0000-0001-9688-8208

M. Peces http://orcid.org/0000-0003-2522-7490

T. B. N. Jensen http://orcid.org/0000-0001-5815-5468

M. Sereika http://orcid.org/0000-0001-7568-1080

M. K. D. Dueholm http://orcid.org/0000-0003-4135-2670

P. H. Nielsen http://orcid.org/0000-0002-6402-1877

## FUNDING

| Funder | Grant(s) | Author(s) |
| --- | --- | --- |
| Villum Fonden | 13351 | P. H. Nielsen |
| Horizon 2020 Framework Programme | 952908 | P. H. Nielsen |

## AUTHOR CONTRIBUTIONS

Z. Kondrotaite, Formal analysis, Investigation, Methodology, Visualization, Writing – original draft, Data curation | J. Petersen, Investigation, Methodology, Writing – review and editing | C. Singleton, Data curation, Methodology, Writing – review and editing | M. Peces, Data curation, Investigation, Writing – review and editing, Methodology | F. Petriglieri, Writing – review and editing, Investigation | T. B. N. Jensen, Data curation, Methodology, Writing – review and editing | M. Sereika, Data curation, Methodology, Writing – review and editing | A. O. H. Daugberg, Data curation, Writing – review and editing | M. Wagner, Conceptualization, Writing – review and editing | M. K. D. Dueholm, Conceptualization, Writing – review and editing, Resources, Investigation | P. H. Nielsen, Supervision, Writing – original draft, Writing – review and editing, Conceptualization, Funding acquisition, Investigation, Project administration

## DATA AVAILABILITY

Data generated and used in this study, metagenomics and metatranscriptomics, are deposited in the EMBL Nucleotide Sequence Database under the BioProject accession number PRJEB67571. Accession numbers for the HQ MAGs are provided in Data S3.

## ADDITIONAL FILES

The following material is available online.

### Supplemental Material

**File S1 (mSystems00322-25-s0001.pdf).** Supplemental figures and tables.
**2undefined (mSystems00322-25-s0003.xlsx).** MAG information.
**3undefined (mSystems00322-25-s0002.xlsx).** Metadata and QC.
**File S4 (mSystems00322-25-s0004.xlsx).** Annotations.

### Open Peer Review

**PEER REVIEW HISTORY (review-history.pdf).** An accounting of the reviewer comments and feedback.

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
