## [Reviewer comments · mSystems]

Ecophysiology and niche differentiation of three genera of polyphosphate accumulating bacteria in a full-scale wastewater treatment plant

Per Nielsen, Zivile Kondrotaite, Jette Petersen, Caitlin Singleton, Miriam Peces, Francesca Petriglieri, Thomas Jensen, Mantas Sereika, Anders Daugberg, Michael Wagner, and Morten Dueholm

Corresponding Author(s): Per Nielsen, Aalborg Universitet

Review Timeline:

Submission Date:	March 7, 2025
Editorial Decision:	April 10, 2025
Revision Received:	July 4, 2025
Accepted:	July 14, 2025

Editor: Jorge Rodrigues

Reviewer(s): Disclosure of reviewer identity is with reference to reviewer comments included in decision letter(s). The following individuals involved in review of your submission have agreed to reveal their identity: Ryuichi Hirota (Reviewer #1)

Transaction Report:

DOI: <https://doi.org/10.1128/msystems.00322-25>

Re: mSystems00322-25 (**Ecophysiology and niche differentiation of three genera of polyphosphate accumulating bacteria in a full-scale wastewater treatment plant**)

Dear Prof. Per Halkjær Nielsen:

Both reviewers were supportive of the work and found the research to be of importance to the mSystems interested community. Reviewers also offered a few suggestions such as elaborating on C utilization profiles to distinguish the three species being considered and a careful review of the supplementary information. I hope you will carefully consider their suggestions when revising the document.

Your manuscript should provide a Data Availability paragraph at the end of the section named Material and Methods. Please, make sure that the data is available to all with appropriate accession numbers.

Revision Guidelines

Sincerely,
Jorge Rodrigues
Editor
mSystems

Reviewer #2 (Comments for the Author):

The work submitted by Kondrotaite et al. proposed to shed more light on the complexity of the presence of PAOs in full-scale wastewater treatment plants (WWTP). They focus on the three types of PAOs known to exist (*Accumulibacter*, *Azonexus* and *Phosphoribacter*) and the differences observed in lab-scale enrichment and WWTP considering their niches. Large-scale data from metagenomics and metatranscriptomics, as well as short-term lab experiments, were used. The information obtained supported previous works but also put forward some new aspects. The paper is important in the thematic field, where it shows the presence and cooperation of different organisms for the same goal and how much we still need to understand processes at full scale. Just a small remark that on lines 349 and 351, the correct figure should be Fig. S8.

This work provides novel insights regarding polyphosphate accumulating organisms (PAOs). The novelty of this study lies in its examination of the ecological and physiological diversity of PAOs in a full-scale wastewater treatment plant (WWTP) using a comprehensive genome-resolved metatranscriptomics approach. Unlike previous studies that have largely focused on *Ca. Accumulibacter* in lab-scale enrichment cultures, this research targets a full-scale environment, offering a more realistic assessment of PAO diversity and function. Additionally, the study includes *Azonexus* and *Ca. Phosphoribacter* alongside *Ca. Accumulibacter*, which is rarely addressed in previous research. Especially the co-existence and niche difference in terms of their carbon-source utilization and electron acceptors are novel important knowledges for full-scale WWTP.

This study was well-designed, involving long-term monitoring of the microbial community in a Danish WWTP over a period of 7–8 years. The researchers systematically analyzed different sections of the WWTP tanks to investigate the responses of resident microorganisms across various operational conditions. I read this manuscript with great interest and believe that it is suitable for publication in *mSystems*. However, I have several concerns and suggestions that should be cleared before publication.

Major points.

1. The differences in the abundance and nutrient niches of the three PAOs are central to the claims of this work. The finding that *Ca. Phospholibacter spp.* are the major species in the WWTP is intriguing; however, it remains unclear whether this population pattern is universal across all WWTPs or specific to the one studied. Can you clarify whether this abundance pattern is a unique characteristic of this particular WWTP or if it is commonly observed in other facilities? Additionally, from the view points of the nutrient niche difference, is this abundance pattern influenced by various factors such as wastewater type (e.g., municipal or industrial) or other specific characteristics of the incoming wastewater at this WWTP? Furthermore, I am particularly interested in the observation that the composition of *Ca. Phospholibacter* increased drastically after January 2019. After this point, the abundance differences became more pronounced than those observed before, suggesting the potential influence of certain factors that may be enhancing the population dynamics. Could you provide insights into what factors might have contributed to this significant shift in abundance?
2. The utilization of glucose, acetate, other carbon sources, and amino acids is one of the key characteristics that distinguishes the physiology and niche differentiation of the three PAOs. Could you elaborate on the evolutionary relationships that led to the emergence of these PAOs? To provide a more comprehensive perspective, I recommend including a broader phylogenetic tree than the one presented in Figure 2. Expanding the tree could help clarify the evolutionary pathways and divergence among these PAOs in relation to their metabolic traits.

3. Several supplementary figures are not cited appropriately in the main text, leading to confusion in understanding the extensive genomic and transcriptomic analyses presented. For example, in line 320, Fig. S6 was mistakenly cited instead of Fig. S5. Please carefully review the entire document to ensure that all supplementary figures and tables are properly cited and accurately referenced in the main text.

4. The authors propose that Pit is the main determinant of PAO characteristics; however, the Pit transporter is known to be a low-affinity Pi transporter. It would be valuable to include a more detailed discussion regarding the characteristics of the Pit transporter and its relationship to PAO phenotypes. Specifically, elaborating on how the low-affinity nature of Pit influences phosphorus acquisition under varying environmental conditions could enhance the interpretation of the findings. Additionally, a phylogenetic analysis of the Pit transporters across a broad range of bacteria may provide more insightful findings and clarify its evolutionary significance in shaping PAO physiology and niche differentiation.

Minor points

L36: Ca. *Azonexus*

L52: *pit*

L61: polyphosphate kinase

L66: P (inorganic phosphate) should be representing as “Pi”.

L101-111 (Methods 2.1): Please include the average value of input P and N amount in the influent wastewater.

L341, L344: Fig S9. They are not cited properly. May be S8?

L353-354: It is difficult to understand this claim “Most extracellular enzymes had low expression in both Ca. *Acumulibacter* and *C. Azonexus*”. I think *Phospholibacter* had low expression than the other two species.

L369-371: Can you add the description regarding the relationship between extracellular Pi concentration and phosphate starvation response of these PAOs?

L440: *ppk1/ppk2-ppa*

Supplementary file4: Please add color indication (heatmap way) to make easily recognize the presence or absence of the gene copies.

Fig. 3: The resolution is too low to recognize each pathway.

Comments to reviewers

Reviewer #1

This work provides novel insights regarding polyphosphate accumulating organisms (PAOs). The novelty of this study lies in its examination of the ecological and physiological diversity of PAOs in a full-scale wastewater treatment plant (WWTP) using a comprehensive genome-resolved metatranscriptomics approach. Unlike previous studies that have largely focused on *Ca. Accumulibacter* in lab-scale enrichment cultures, this research targets a full-scale environment, offering a more realistic assessment of PAO diversity and function. Additionally, the study includes *Azonexus* and *Ca. Phosphoribacter* alongside *Ca. Accumulibacter*, which is rarely addressed in previous research. Especially the co-existence and niche difference in terms of their carbon-source utilization and electron acceptors are novel important knowledges for full-scale WWTP.

This study was well-designed, involving long-term monitoring of the microbial community in a Danish WWTP over a period of 7–8 years. The researchers systematically analyzed different sections of the WWTP tanks to investigate the responses of resident microorganisms across various operational conditions. I read this manuscript with great interest and believe that it is suitable for publication in *mSystems*. However, I have several concerns and suggestions that should be cleared before publication.

Major points.

1. The differences in the abundance and nutrient niches of the three PAOs are central to the claims of this work. The finding that *Ca. Phospholibacter spp.* are the major species in the WWTP is intriguing; however, it remains unclear whether this population pattern is universal across all WWTPs or specific to the one studied. Can you clarify whether this abundance pattern is a unique characteristic of this particular WWTP or if it is commonly observed in other facilities? Additionally, from the view points of the nutrient niche difference, is this abundance pattern influenced by various factors such as wastewater type (e.g., municipal or industrial) or other specific characteristics of the incoming wastewater at this WWTP? Furthermore, I am particularly interested in the observation that the composition of *Ca. Phospholibacter* increased drastically after January 2019. After this point, the abundance differences became more pronounced than those observed before, suggesting the potential influence of certain factors that may be enhancing the population dynamics. Could you provide insights into what factors might have contributed to this significant shift in abundance?

Answer: As the reviewer mentions, the species of *Ca. Phosphoribacter* were the most abundant PAO in the WWTP analyzed in this study. This is a general observation in Danish EBPR plants (Singleton et al, 2022). The global analysis we made in 2022 (Dueholm et al., 2022) shows that

species in *Ca. Phosphoribacter* are also the most abundant PAO in most EBPR plants globally. To make it clearer in the paper, additional explanation was added in lines 224-226.

The abundances of the different PAOs are influenced by various factors such as process design, wastewater type, operation, immigration via wastewater and interactions. We do not have any specific explanations for the observed dynamics and have in other studies tried to understand these dynamics (Peces et al., 2022), but without success. We have, however, in an ongoing study found that a specific species from the *Patescibacteria* seems to predate on members of *Ca. Phosphoribacter*, and that can likely explain such abrupt abundance changes. We hope to have a paper ready on this topic later this year. We have included a bit more about potential factors controlling the abundances (lines 215-218).

2. The utilization of glucose, acetate, other carbon sources, and amino acids is one of the key characteristics that distinguishes the physiology and niche differentiation of the three PAOs. Could you elaborate on the evolutionary relationships that led to the emergence of these PAOs? To provide a more comprehensive perspective, I recommend including a broader phylogenetic tree than the one presented in Figure 2. Expanding the tree could help clarify the evolutionary pathways and divergence among these PAOs in relation to their metabolic traits.

Answer: We appreciate the reviewer's insightful suggestion regarding the evolutionary relationship among the three PAOs and their metabolic differentiation. A broader phylogenetic tree encompassing a wider range of related taxa has been included in Supplementary Fig. S1, including all the MAGs generated in this study. The more comprehensive view of the evolutionary divergence of the PAOs (Fig. S1) highlights the distinct lineages of the three PAOs and thus support their metabolic differentiation pattern (*Ca. Accumulibacter* and *Azonexus* being more closely related, while *Ca. Phosphoribacter* belongs to a different phylum). To clarify this point, we have added the reference to the supplementary figure in the main text in line 512-515.

3. Several supplementary figures are not cited appropriately in the main text, leading to confusion in understanding the extensive genomic and transcriptomic analyses presented. For example, in line 320, Fig. S6 was mistakenly cited instead of Fig. S5. Please carefully review the entire document to ensure that all supplementary figures and tables are properly cited and accurately referenced in the main text.

Answer: We have checked and changed, where necessary, all the references to the supplementary figures.

4. The authors propose that Pit is the main determinant of PAO characteristics; however, the Pit transporter is known to be a low-affinity Pi transporter. It would be valuable to include a more

detailed discussion regarding the characteristics of the Pit transporter and its relationship to PAO phenotypes. Specifically, elaborating on how the low-affinity nature of Pit influences phosphorus acquisition under varying environmental conditions could enhance the interpretation of the findings. Additionally, a phylogenetic analysis of the Pit transporters across a broad range of bacteria may provide more insightful findings and clarify its evolutionary significance in shaping PAO physiology and niche differentiation.

Answer: A comparison between *pit* and *pstASCB* transport system is added in lines 54-56. While the phylogenetic analysis of the *pit* transporter across a broad range of bacteria is an interesting idea and may provide more insight into the PAOs phenotype, we find that it falls outside the scope of this paper, and it would require a comprehensive in-depth analysis deserving its own paper. In the present study we have focused on three well-known PAOs genera and their niche differentiations and not what defines the PAO phenotype.

Minor points

L36: *Ca. Azonexus* – has been changed.

L52: *pit* – has not been changed as we talk about the transport system and not the gene.

L61: polyphosphate kinase – has been changed.

L66: P (inorganic phosphate) should be representing as “Pi”. – we prefer to use P for phosphate and have done it consistently in the paper for various forms of phosphate. We have clarified that P is for phosphate in line 42.

L101-111 (Methods 2.1): Please include the average value of input P and N amount in the influent wastewater. – The values were added in lines 114-115.

L341, L344: Fig S9. They are not cited properly. May be S8? – has been changed.

L353-354: It is difficult to understand this claim “Most extracellular enzymes had low expression in both *Ca. Acumulibacter* and *C. Azonexus*”. I think *Phospholibacter* had low expression than the other two species. – it has been changed to make it more clear.

L369-371: Can you add the description regarding the relationship between extracellular Pi concentration and phosphate starvation response of these PAOs? – since we have no experimental evidence of such relationship it would be only speculations, so we have decided not to include it.

L440: *ppk1/ppk2-ppa* – has been changed.

Supplementary file4: Please add color indication (heatmap way) to make easily recognize the presence or absence of the gene copies. – has been changed.

Fig. 3: The resolution is too low to recognize each pathway. –The original file is fine, it is just in the world file.

Reviewer #2 (Comments for the Author):

The work submitted by Kondrotaite et al. proposed to shed more light on the complexity of the presence of PAOs in full-scale wastewater treatment plants (WWTP). They focus on the three types of PAOs known to exist (*Accumulibacter*, *Azonexus* and *Phosphoribacter*) and the differences observed in lab-scale enrichment and WWTP considering their niches. Large-scale data from metagenomics and metatranscriptomics, as well as short-term lab experiments, were used. The information obtained supported previous works but also put forward some new aspects. The paper is important in the thematic field, where it shows the presence and cooperation of different organisms for the same goal and how much we still need to understand processes at full scale. Just a small remark that on lines 349 and 351, the correct figure should be Fig. S8.

Thank you for nice words. The correct figure is now inserted.

Re: mSystems00322-25R1 (**Ecophysiology and niche differentiation of three genera of polyphosphate accumulating bacteria in a full-scale wastewater treatment plant**)

Dear Prof. Per Halkjær Nielsen:

Your manuscript has been accepted, and I am forwarding it to the ASM production staff for publication. Your paper will first be checked to make sure all elements meet the technical requirements. ASM staff will contact you if anything needs to be revised before copyediting and production can begin. Otherwise, you will be notified when your proofs are ready to be viewed. Before the manuscript can be published, ASM requires that all manuscripts reporting novel sequence data to provide a short paragraph at the end of Material and Methods section with information on the data repository and accession numbers. Please, make sure to comply with this requirement to avoid any delay on the publication.

Thank you for submitting your paper to mSystems and for allowing me to serve as Editor for this important research work.

Sincerely,
Jorge Rodrigues
Editor
mSystems